# The Art of Medical Diagnosis: Lessons on Interpretation of Signs from Italian High Renaissance Paintings

**DOI:** 10.3390/diagnostics15030380

**Published:** 2025-02-05

**Authors:** Marcin Śniadecki, Anna Malitowska, Oliwia Musielak, Jarosław Meyer-Szary, Paweł Guzik, Zuzanna Boyke, Martyna Danielkiewicz, Joanna Konarzewska, Cynthia Aristei

**Affiliations:** 1Department of Gynaecology and Obstetrics, Medical University of Gdańsk, 80-210 Gdańsk, Poland; mmdanielkiewicz@gumed.edu.pl; 2Department of Ethics, Adam Mickiewicz University, 61-712 Poznań, Poland; anna.malitowska@amu.edu.pl; 3Department of Surgical Oncology, Medical University of Gdańsk, 80-210 Gdańsk, Poland; omusielak@gumed.edu.pl; 4Department of Paediatric Cardiology and Congenital Heart Defects, Medical University of Gdańsk, 80-210 Gdańsk, Poland; jaroslaw.meyer-szary@gumed.edu.pl; 5Department of Gynaecology and Obstetrics, City Hospital Rzeszów, 35-055 Rzeszów, Poland; pawelguzik@poczta.onet.pl; 6Department of Art History, University of Gdańsk, 80-308 Gdańsk, Poland; z.boyke.073@studms.ug.edu.pl; 7Department of Radiology, University of Gdańsk, 80-214 Gdańsk, Poland; mijo@gumed.edu.pl; 8Department of Medicine and Surgery, University of Perugia and Perugia General Hospital, 06129 Perugia, Italy; cynthia.aristei@unipg.it

**Keywords:** breast cancer, diagnosis, diagnostic methodology, methodological competence, sign interpretation, Renaissance paintings

## Abstract

Medicine is struggling with the constantly rising incidence of breast cancer. The key to this fight is to be able to speed up diagnosis, as rapid diagnosis reduces the number of aggressive or advanced cases. For this process to be effective, it is necessary to have the right attitude toward diagnosis as a research practice. Our critical analysis of diagnosis, as a methodology of medical science, reflects on it as a research practice that is regulated in a socio-subjective way by a methodological culture. This position allows us to contrast critical methodological culture with the habitual–practical, or methodical, culture of practicing diagnosis. We point to the interpretative status of medical analyses performed by medical historians by referring to Italian Renaissance paintings and historical–artistic interpretations. In this field, analyzing disputes between researchers as a clash of methodologies in the ways interpretation transforms signs into meaning is a critical methodological reflection. Medicine is a diverse scientific discourse with a paradigmatic structure in which new ways of conducting diagnostic tests may appear. It is only possible to see this from the methodological level. In addition, passive respect for existing patterns of conduct hinders an exchange of views between researchers, which limits the possibility of correcting research procedures. The ultimate consequence of such passivity is an inability to improve diagnosis, which, in turn, harms the interests of patients. In this regard, it is worth remembering that the paramount objective of diagnosis is not the disease, but the patient.

## 1. Introduction

Breast cancer is the most frequently diagnosed malignant tumor in the world across the entire general population, with the female gender being the most at-risk group [1,2]. At present, the latency period of breast cancer is becoming shorter as the onset and quantity of the risk factors are increasing [3,4]. The progressive incidence of breast cancer necessarily brings about a rise in the attention paid to an early diagnosis [4,5,6,7]. When early testing and diagnosis take place, the percentage of aggressive or advanced cancer cases decreases, and the number of early-stage or low-grade cancer detections increases [6,8]. Patients go to doctors to be examined because the assumption is that doctors have the knowledge and tools and have mastered the art of interpreting symptoms and signs well enough to make a diagnosis. In breast cancer, the information from visual signs is extremely important, since diagnosis of this cancer is mainly image-based [9]. Therefore, we can advance the thesis that approaches to image interpretation play a key role in today’s medicine; consequently, it is worth asking appropriate questions about the role of interpretation.

Since diagnostic tests are so important in the fight against a pressing global public health problem [10], it is necessary to have the right attitude toward diagnosis as a research practice. This attitude can be summarized as involving a quartet of tasks: (1) an analysis and reconstruction of the concept of diagnosis, and of its related standards and procedures; (2) criticality toward the diagnostic process using cognitive criteria, including ethical ones; (3) the structured development of evidence-based proposals for solutions to problem situations; and (4) pragmatics, i.e., a cooperative stance toward other knowledge disciplines and institutions in order to ensure conditions for effective implementation and application of the proposed solutions in medical practice.

At the center of our interest is how each diagnosis as a purposeful, methodical, and iterative process flows from the development of the disease or the new results obtained from any additional tests undertaken. By iterative, we mean that it is subject to constant evaluation, re-evaluation, and verification as new information emerges. The quality and effectiveness (performance) of diagnostic tests depends on the degree to which they are developed and implemented into broad use, as well as recognized, understood, accepted by practitioners, and applied in individual cases.

These tasks belong to the field of methodology of science, a discipline investigating research practice with regard to the activities it comprises and their results. As such, methodology addresses both research objectives and the instructions on how these objectives should be achieved by researchers. More precisely, methodology reconstructs research procedures as they are employed in practice. However, its tasks, apart from reconstruction, also include the evaluation of investigation procedures as to their adequacy to the goals of a given discipline.

Thus, being competent in the discipline’s research and examination methods is not only about the ability to apply them correctly, i.e., in accordance with the state of the art, but also the ability to reflect on their validity and purposefulness. The above context encourages deepening methodological reflection on diagnosis and developing a reflective and critical attitude toward medical research practice.

## 2. Methodological Competence

Knowledge of methods, so methodological competence, relies on the acceptance of beliefs that define the goals of research activities and characterize the activities required to achieve these goals. Since medicine, both as a scientific discipline and as medical practice, is a social phenomenon, institutionalized and regulated by professional standards, the rules of medical conduct identified by the methodology are shared and recognized as valid by medical professionals. Accepting these rules means that they determine both the objectives of professional conduct and the means of achieving them.

The rules could be reconstructed in the form of theories relating to individual stages of the research process, e.g., theories of the interpretation or generalization of research results, or knowledge about research models, thus constituting a set of rules for using specific models in the process of empirical analysis and verification of research results.

There are two attitudes in which methodological competence can be exercised. In the first one, methodological competence is application-oriented and refers to the ability of a practitioner to follow the methodological rules suggested by the state of the art, i.e., to apply a procedure or rule in a manner appropriate to given circumstances. In the second attitude, methodological competence entails consideration of the performance of applied research rules, enabling questions on their presuppositions, efficiency, and effects of the conduct resulting from the rule acceptance. In that case, methodological competence is reflection-oriented.

Knowledge of methods, while shared within a professional community, does not have to be explicit. What is typical of the application-oriented attitude is that being acquainted with the research rules does not necessarily mean the ability to precisely verbalize them or—even more so—to justify them theoretically. It does not mean that the application-oriented attitude excludes any overt thought process. Far from it, a diagnostician as a researcher makes decisions and implements them in a dynamic thought process based on the assessment of the situation, anticipating consequences, etc.

In summary, it can be said that reflective methodological competence is more than just a specialized research competency.

## 3. The Art of Diagnosis

When characterizing the concept of methodological competence, we will refer to the controversy surrounding the so-called medical diagnosis of High Renaissance (*Cinquecento*, 1500–1599) paintings as allegedly depicting breast cancer. These discussions were published in The Lancet and The Lancet Oncology. Examples include the articles “The portrait of breast cancer and Raphael’s La Fornarina” and “Earliest evidence of malignant breast cancer in Renaissance paintings” [11,12]. Considerations of Jonathan K. Nelson’s “Cancer in Michelangelo’s Night. An analytical framework for retrospective diagnoses” [13] forms a critical contribution to this discussion.

According to the Merriam-Webster dictionary, (medical), *diagnosis* is the art or act of identifying a disease from its signs and symptoms [14]. What is the art of diagnosis? It is an art in a technical sense, i.e., actionable knowledge of rules or a capability to act in accordance with them. When dealing with signs, diagnosis is an art or activity of interpretation.

The interpreter is then someone who operates on signs, transforming them, so that eventually, after many transformations, they provide an answer to a question, because interpreting as a cognitive (research) activity is intentional. Furthermore, diagnosis is an iterative process in which the diagnosis is continually revised as new information becomes available, resulting from the development of the disease or as new results from additional tests become available.

Interpreting as the transformation of signs is not arbitrary but methodical. We have ways to specify the criteria for correct interpretation and make them intersubjective. In medicine, more precisely, in radiological examinations of the mammary gland, interpretations of visual data, such as a sequence of magnetic resonance images (MRI), are regulated by standard nomenclature and diagnostic criteria, as for instance in the use of the Breast Imaging Reporting and Data System (BI-RADS), which serves as a specialized dictionary [15]. Among other things, it defines what vocabulary should be used in the interpretation of the image. For example, the word “ovoid” cannot be used in the description of a region of interest (ROI) because BI-RADS interpretation guidelines specify the use of the word “oval” to denote a mass that has a rounded and slightly elongated shape (including lobulated), while “ovoid” is 3-D, and “oval” is 2-D [16]. The guidelines for the correct interpretation of an image, of course, are not limited to the use of appropriate terminology to describe a ROI but also include diagnostic knowledge. For example, it is assumed that the correct interpretation of ductal enhancement occurring two minutes after contrast administration should be stated as a “suspicious enhancement”, since in most cases it is a typical indicator of invasive ductal carcinoma [17].

Interpretation is an essential element of any research process; as a consequence, even seemingly objective scientific activities, such as description or analysis, are derivatives of interpretation or even its direct variants.

Meanwhile, when the authors of the articles mentioned earlier, make a medical diagnosis of the elements in Raphael’s *La Fornarina*, Tosini’s *the Night*, or Maso di San Friano’s *The Allegory of Fortitude*, they separate their medical analysis from their interpretation of art historical significance [11,12]. Moreover, they distinguish medical analysis itself from artistic analysis, assuming the possibility of contact with an image, which is completely independent of the act of experiencing it. In other words, they ensure the possibility of adopting a pre-esthetic and research attitude in which the essential properties of the work are recognized, completely independent of an experience of its esthetic properties. *The Night* was based on a sculpture by Michelangelo located in Sacrestia Nuova, San Lorenzo church, Florence. The model from *The Night*, is a masculine-looking female form. Though it was difficult before 1500 for Italian artists to find female models for nude studies, it was not impossible, and certainly after 1500, it became much easier. In the Michelangelo figure, it is quite possible that the appearance of breast cancer was overlaid (with the breast) onto a male figure based on Michelangelo’s observation of a nude female model.

In both cases, *The Night* and *The Allegory of Fortitude*, however, we are dealing with recognizing meaning based on signs. Medical diagnosis using images involves a transformation of signs just as artistic analyses do. Both are interpretations, and the differences between them do not result from the characteristics of the subject of interpretation but from the particular methods adopted. The same image can be interpreted in many ways, depending on the cognitive needs that define the problem the interpreter is facing.

Undoubtedly, the authors of medical diagnoses of the figures in the paintings interpret the properties of the depicted breast as indexical signs. Such signs are used, for example, by the authors of the “Earliest evidence of malignant breast cancer in Renaissance paintings”, interpreting paintings that are—this is their thesis—“two proposals of the earliest pictorial representations of breast cancer dated to the 16th century” [12]. In Peirce’s semiology, an indexical sign (indicator or sign) would be characterized by the fact that there is a real, physical connection between the signifier and the signified [18]. Peirce’s typology considers three types of signs: index, icon and symbol. An iconic sign represents an object if the sign and the object are similar. Thus, a particular arrangement of color spots on the canvas or the shape given to a block of stone by the sculptor may represent a woman’s breast because they are more or less similar to a woman’s breast. Finally, in the case of symbols, signs represent objects when there is a convention that assigns denotations to them, which means that symbols require interpretation based on specific conventions. For example, in the painting *La Fornarina*, the naked left breast interpreted as naked, can be a symbol of beauty, truth, and goodness (following the classicist convention), whereas in the Christian tradition, the left breast can be seen as a symbol of sin (Figure 1).

*La Fornarina* supports her breast with one hand and covers her womb with the other—this pose resembles the ancient model of *Venus pudica*, i.e., a modest, shy Venus: a gesture of modesty that yet directs the viewer’s gaze to what she actually seeks to conceal. However, in the context of Raphael, mention should be made of Titian’s Renaissance work, *Sacred and Profane Love* (1515–1516), depicting two women, naked and clothed (https://www.collezionegalleriaborghese.it/en/opere/sacred-and-profane-love (accessed on 1 December 2024). It seems that the nude figure symbolizes the goddess Venus (Venere Celeste). It has the features of Hellenistic statues. In a woman in robes, we can see ’earthly’ Venus—(Venere Vulgare) or Medea as a symbol of momentary happiness. In this symbolic context, the covering of the left breast by an earthly woman with the appearance of a goddess, whom art history claims was the painter’s lover (Margherita Luti), is an element of the erotic love game between the artist and his muse. The covering gesture is significant as consent and an invitation to sin together, not what is covered. Maybe there is a healthy breast behind the curtain, or perhaps it has cancer—it does not matter here.

We can try to look at another High Renaissance painting, i.e., *The Rape of the Sabine Women* by Giovanni Antonio Bazzi (known as Sodoma), through the prism of all three types of signs (Figure 2).

On the left side of the painting *The Rape of the Sabine Women*, three men hold a woman. She is only half covered by a red robe, and her bare breasts are the focus. Their visible disproportion and deformity may indicate the earliest representation of a tumor of this gland in the Renaissance. The colors also suggest that we are looking at a woman afflicted with some disease—the left breast is slightly darker than the right, and a distinct chiaroscuro was used to model it. Moreover, the left breast is not only smaller, but one can also see a pull in the skin, perhaps even despite the hair covering this part of the breast. A similar impression of pathology, as in the model on the left, can be seen in the left breast of the second model—painted closer to the center of the painting (Figure 3A,B).

The Roman story of rape, or abduction, was a popular subject during the Renaissance. In 15th-century Italy, which was drawing cultural strength and much of its iconography from its Roman roots, the abduction was often depicted on *cassoni* (marriage chests) and later, in larger paintings, such as Sodoma’s, as symbolic of the importance of marriage for familial and cultural continuity. The iconography and symbolism are consistent with the interpretation of representations of the Sabine women’s bodies, especially their bare breasts (but also the red cover symbolizing love and sin) in terms of indexical signs of disease, signifying something deviating from the norm, and contrary to nature.

Bearing in mind the realism of High Renaissance paintings, the presented deformation of the nipple or the darker color of slightly retracted skin, can be considered as symptoms (indexical signs) of cancerous changes. Similarly, as in the case of *The Night* or *Allegory of Fortitude*, we could look at another work by Sodoma, entitled *The Fall of Phaeton* (https://worcester.emuseum.com/objects/6465/the-fall-of-phaeton (accessed 1 December 2014), in which one could see the same model as in *The Rape of the Sabine Women*—it can be assumed that there is a scar on the left breast. It is also worth mentioning that many, if not most, High Renaissance paintings depicted the symptoms of left-sided breast pathology.

Reflective methodological competence would, therefore, require that research on works of art suspected of representing breast cancer over the centuries be compared with the state of medical knowledge specific to a given era and examine how medical discourse relates to artistic practice. They could also be combined with biographical information that would link the figure depicted in the painting (e.g., *La Fornarina*) with a real person (Margherita Luti), and preferably also with the history of her illness. From a methodological point of view, it may be interesting to know how and why the interpretations provided by medical science differ from those present in the history of art.

The study of images—from medical to pictorial—requires research itself. According to Gillian Rose, reflecting on the methods that can be used to interpret visual images, is a task of critical visual methodology [19]. The methodological approach considers the cultural conditions of interpretation, in particular the fact that the beliefs respected and accepted by the members of a given cultural community determine the ways of encoding and decoding signs they use. Scholars who are aware that what is seen and how it is seen are culturally constructed, consider their own way of looking at images.

## 4. Threats Posed by Limited Methodological Competence and Ways to Cope

Our analogy between historical and artistic interpretations points to the interpretive status of medical analyses performed by medical historians. The disputes that are possible between them are disputes between methods of interpretation. A critical examination of these is possible but at the level of critical methodological reflection.

Discoveries of the history of medicine lack response from other sciences. Art historians, in turn, ignore medical studies that comment on art. The absence of dialog between such different disciplines, even though they refer to the same subject, is the result of the limitations of the dominant, application-oriented methodological competence.

Tacit respect for existing patterns of conduct hinders the exchange of views between researchers, which, in turn, limits the possibility of correcting research procedures. In the process, diagnosticians who have been trained to solve problem situations intuitively fail to notice or accept innovative ways of conducting research. Only at the methodological level is it clear that medicine is a diverse scientific discourse with a paradigmatic structure in which new ways of conducting diagnostic tests may appear.

Where verbal behavior is habitual and practically oriented, methodological experience is characterized by argumentative and persuasive weakness. Application-oriented methodological competence means that researchers are unable to conduct interpretative disputes competently, or at a level that goes beyond disputes conducted in a habitual and practical manner. Dispute resolutions occur as technical arguments, or in the worst cases, as non-cognitive and non-scientific arguments, used to defend the intuitively and habitually held scientific worldview.

A reflective methodological competence of diagnosticians is in the interest of patients. A diagnostician is an expert decision-maker who undertakes empirical research for practical and social reasons in order to solve patients’ health problems. Certain of these health problems can only be noticed and solved from a critical methodological perspective. To this extent, the definition of diagnosis as identifying a disease is insufficient. It omits the carrier/subject of the disease. The object of diagnosis is the disease, but the subject is the patient.

Diagnostic methods today are based on the medical analyses of past researchers, selecting a basic framework as a “standard patient” of every particular disease. A critical approach should be integrated into diagnosing a patient, taking into account evidence-based medicine as well as the holistic approach of physical examination and history. Interpretation in medicine has been with us since anatomy lessons. The medical profession is also limited by our context in how we diagnose. For example, the notion that “doctors are never wrong” is a taboo topic, which gives us restrictions in terminology or interpretation they can lead to a missed diagnosis or rather an unwillingness to search further than the “most common diseases”, applying the rules of proper differential diagnosis.

Building upon this perspective, it becomes evident that the endeavor to apply contemporary diagnostic standards to historical artworks not only presents a formidable challenge but also underscores the vast chasm between past and present medical practices. The absence of direct patient history, or anamnesis, is a significant barrier. In the realm of modern medicine, the patient’s history is foundational to diagnosis, offering insights into symptoms, onset, and progression of the condition, as well as personal and familial health history that might influence the diagnosis. Historical artworks, however, provide no such narrative, leaving much to conjecture and interpretation based on visible signs alone.

Moreover, the lack of access to what is presently recognized as multimodal imaging further compounds this challenge. Today, diagnostics is immensely reliant on a variety of imaging techniques—ranging from X-rays and MRIs to CT scans and ultrasounds—that provide a multidimensional view of the patient’s internal state. These modalities allow for a comprehensive assessment of the disease, facilitating a more accurate and nuanced diagnosis. When examining historical artworks, such diagnostic tools are, of course, unavailable, and any attempt to understand the medical conditions depicted is limited to the visual cues present in the artwork itself.

This limitation is not merely technical but also interpretative, as the understanding of symptoms, diseases, and their representations has evolved significantly over time. What was once considered a sign of a particular condition may now be understood as indicative of another, or perhaps recognized as a normal variation. Furthermore, the symbolic and cultural significance of certain motifs and depictions in art may lead to misinterpretation when viewed through a purely medical lens.

Thus, while historical artworks can provide fascinating insights into the medical knowledge, practices, and prevailing health conditions of past eras, they also highlight the challenges inherent in bridging the gap between historical and contemporary medical diagnostics [20,21]. This endeavor necessitates a multidisciplinary approach that combines art historical analysis with medical expertise, acknowledging the limitations and biases inherent in both fields [22]. Only through such a comprehensive and nuanced approach can we hope to approximate an understanding of the health and diseases of the past, recognizing that our interpretations are inevitably shaped by the lens of contemporary medical knowledge and technology.

In recent years, Artificial Intelligence (AI) has become increasingly important in diagnostics. When examining contemporary imaging methods that incorporate AI, we can indeed draw parallels to artists’ efforts in creating images that might serve diagnostic purposes [23]. Artificial intelligence is revolutionizing the landscape of breast cancer diagnosis, promising enhanced accuracy, efficiency, and personalized patient care. By leveraging machine learning algorithms, AI systems can analyze vast datasets of mammographic images, identify patterns, and detect anomalies with a precision that often surpasses human capability. For instance, deep learning models have demonstrated high sensitivity in identifying early-stage malignancies, even in dense breast tissues where traditional methods may falter. Furthermore, AI aids in the reduction in false positives and negatives, streamlining the differential diagnosis process by integrating multimodal data, including imaging, genomic profiles, and patient history [24,25]. This transformative approach not only accelerates the diagnostic timeline but also reduces the cognitive burden on clinicians, fostering more informed decision-making. As AI technologies continue to evolve, their integration into diagnostic workflows is poised to shift the paradigm from reactive to proactive medicine, emphasizing early intervention and individualized treatment pathways.

## 5. Cover Story and Interpretational Approach

A 44-year-old female patient, came to a senologist (breast specialist) for a consultation after she remarked a swelling, redness, and lump in the upper, outer quadrant of her right breast (Figure 4a,b). She had never been pregnant, she had a history of breast cysts, and her family history of cancer was negative. The patient is a physiotherapist by profession and denied breast injury. She was first referred to her family doctor, who, after examining her breasts, scared her that she had cancer and immediately referred her to a breast cancer unit. However, the patient wanted to verify it as soon as possible and made a private office appointment with a senologist.

On palpation, there was an isolated resistance of approximately 5 cm in size and it was tender. The retracted nipples were on both sides. The senologist, after performing a breast ultrasound examination, categorized the lesion as BI-RADS-4b (Figure 4c–e), and ordered a biopsy, but after performing an MRI examination due to the dense, glandular structure of the breasts (Figure 4f).

The histopathology showed, after considering the clinical and radiological picture, the presence of immature fibrous tissue with resorption and reparative changes after previous damage, in the form of sparse resorptive histiocytic granulations and cholesterol remnants. The ducts and lobules were changed reactively. The diagnosis concluded was a consequence of trauma or damage to the cyst wall. The histopathological picture is consistent with the diagnosis of idiopathic granulomatous mastitis. This inflammation typically affects women aged 20–40, pregnancy is a risk factor, in 20% of cases the inflammation is bilateral, and in half of the cases, it clinically resembles inflammatory cancer. In ultrasound, it usually presents an area of hypoechoic foci up to 80 mm [26].

The cause of these clinical symptoms was most likely an injury unnoticed by the patient (perhaps during her physiotherapeutic work). Nine months after that event, the patient did not report any complaints.

In the symbolic and iconic signs spectrum is the patient’s medical history and awareness of what a breast cancer diagnosis means to the patient, both professionally and socially. However, an indexical sign for breast cancer is questionable even in real patients, so considering the interpretation of paintings—how we can diagnose pictures instead of a patient? Diagnosis requires all the senses, knowledge, personal experience, and research methods, and the diagnosing is not a shortcut—this is methodological competence.

## 6. Conclusions

Both fine art images and medical images require interpretation by the viewer. The diagnostic process is a piece of art looking at the patient and his acts holistically. Interpretation is the basis of medical diagnosis and requires a methodological attitude. Medicine is not a matter of correctness but context, and evidence-based science makes interpretation objective and more contextual.

## Figures and Tables

**Figure 1 diagnostics-15-00380-f001:**
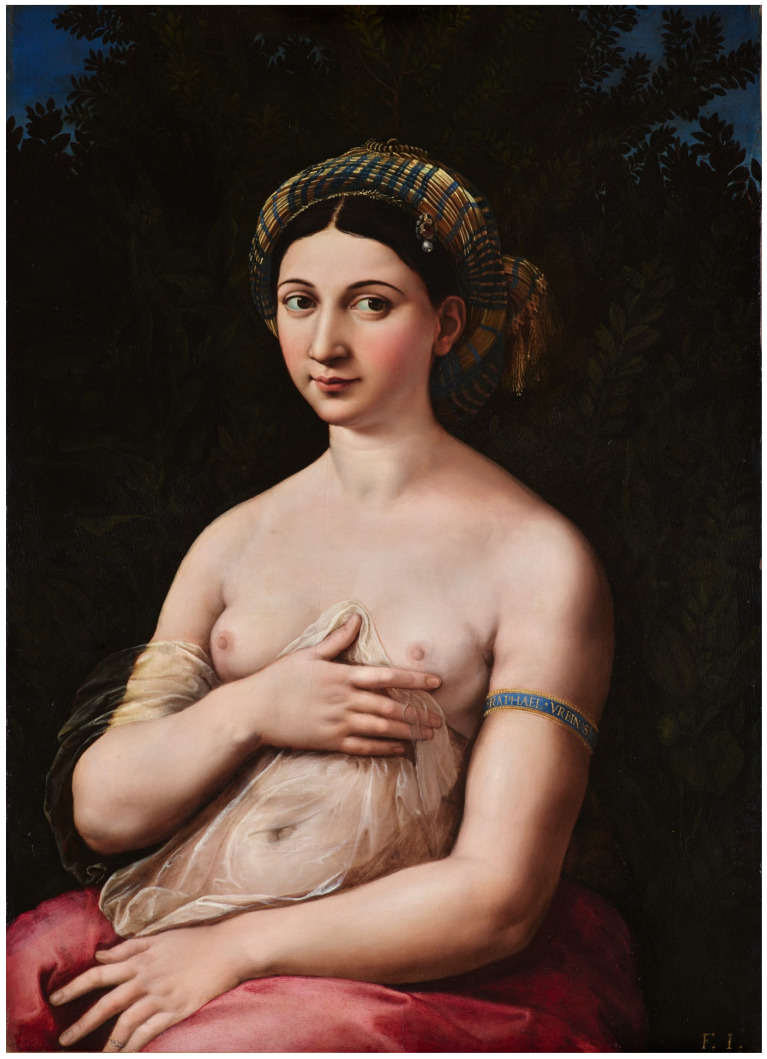
*La Fornarina*, A Portrait of a Young Woman (1518–1519), oil on wood panel, Raffaello Sanzio da Urbino (1483–1520). In: Galleria Nazionale d’Arte Antica in Rome (reproduced with permission).

**Figure 2 diagnostics-15-00380-f002:**
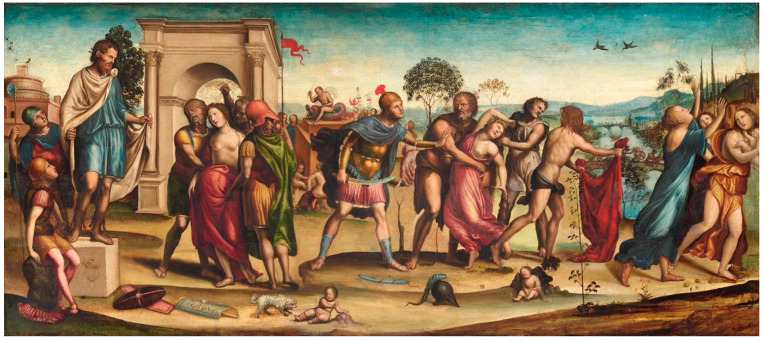
*The Rape of the Sabine Women* (1505–1507), oil on wood panel, Giovanni Antonio Bazzi detto Sodoma (1473–1549). In: Galleria Nazionale d’Arte Antica in Rome (reproduced with permission).

**Figure 3 diagnostics-15-00380-f003:**
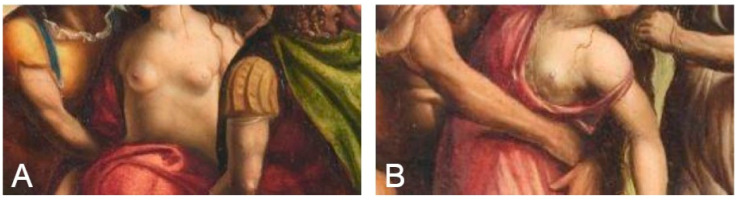
*The Rape of the Sabine Women* (1505–1507), oil on wood panel, Giovanni Antonio Bazzi detto Sodoma (1473–1549). In: Galleria Nazionale d’Arte Antica in Rome. (**A**) The breasts of a model on the left side of the painting, (**B**) The left breast of a model closer to the center of the painting (reproduced with permission).

**Figure 4 diagnostics-15-00380-f004:**
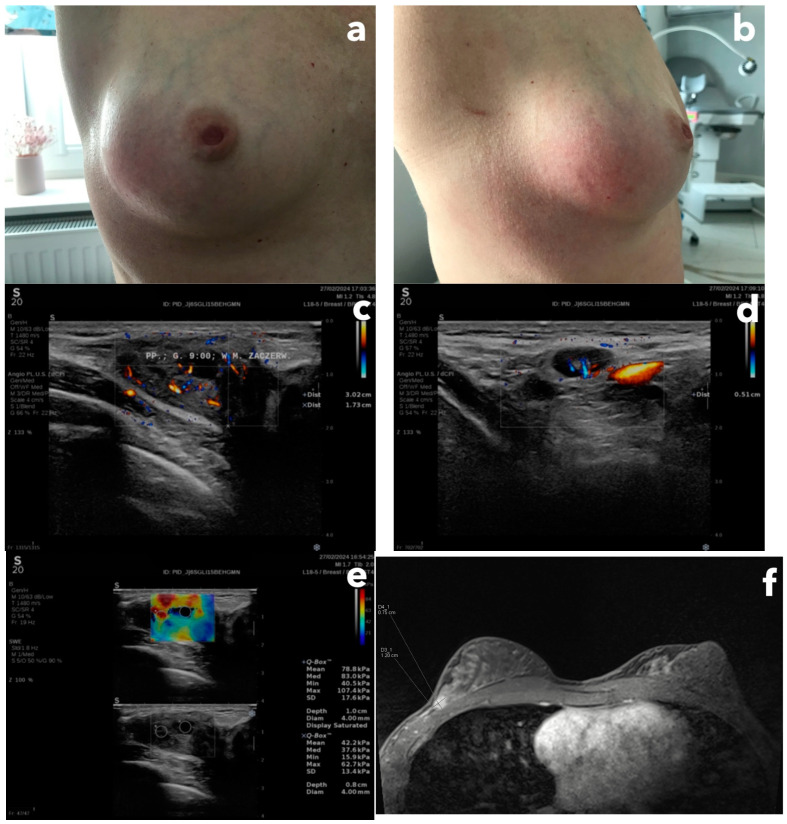
(panel). (**a**) Front view at the right breast with its distortion, swelling, and redness of the skin, (**b**) Lateral view at the right breast, (**c**) B-mode ultrasound image with microflow presentation, corresponding to the redness location, BI-RADS-4b, (**d**) a suspected axillary lymph node with enlarged cortex, (**e**) shear-wave elastography presentation of the lesion showing the increased stiffness (mean stiffness value equals 78.8 Kilopascals) (**f**) T1-weighted MRI image showing rapid contrast enhancement with diffusion restriction of the lesion localized at 7/8 h, subcutaneously, and in adherence to the fascia of the pectoralis major muscle.

## Data Availability

No new data were created or analyzed in this study.

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
