# Peer review of "The Art of Medical Diagnosis: Lessons on Interpretation of Signs from Italian High Renaissance Paintings"

_diagnostics, 2025, doi:10.3390/diagnostics15030380_

Round 1
Reviewer 1 Report
Comments and Suggestions for Authors
I would like to thank the authors for the article with the most different perspective I have evaluated in a long time. In the article that deals with art and medicine in the same article, they have presented their perspectives on the diagnosis of diseases, and have reflected the change from the past to the present. There is no aspect to criticize. In the future, artificial intelligence will be used as an important diagnostic tool, and I recommend that they share the results of the tables evaluated in the article when evaluated with AI in order to provide an even different perspective to the article. My minor suggestion regarding the article is to share more up-to-date GLOBOCAN data.
Best regards
Author Response
We would like to thank you a lot for your time spent on reviewing our manuscript and your valuable comments.
We have found two comments requiring response:
Comment 1
"In the future, artificial intelligence will be used as an important diagnostic tool, and I recommend that they share the results of the tables evaluated in the article when evaluated with AI in order to provide an even different perspective to the article."
Response 1
Thank you very much for your suggestion, but we must apologize, as we are going to submit another article including AI topic, with the working title "Intersections of Art and Diagnostics in Breast Cancer Detection: A Pictorial Essay Focusing on Self-Imaging" (a submission of this article is planned to one of the journals devoted exclusively to ultrasound).
However, in order to propose some form of response and to additionally signal the new context of the work, we would like to supplement the current text with the following (in the discussion section):
In recent years, Artificial Intelligence (AI) has become increasingly important in diagnostics. When examining contemporary imaging methods that incorporate AI, we can indeed draw parallels to Raphael's efforts in creating images that might serve diagnostic purposes [20]. Artificial intelligence is revolutionizing the landscape of breast cancer diagnosis, promising enhanced accuracy, efficiency, and personalized patient care. By leveraging machine learning algorithms, AI systems can analyze vast datasets of mammographic images, identify patterns, and detect anomalies with precision that often surpasses human capability. For instance, deep learning models have demonstrated high sensitivity in identifying early-stage malignancies, even in dense breast tissues where traditional methods may falter. Furthermore, AI aids in the reduction of false positives and negatives, streamlining the differential diagnosis process by integrating multimodal data, including imaging, genomic profiles, and patient history [21,22]. This transformative approach not only accelerates the diagnostic timeline but also reduces the cognitive burden on clinicians, fostering more informed decision-making. As AI technologies continue to evolve, their integration into diagnostic workflows is poised to shift the paradigm from reactive to proactive medicine, emphasizing early intervention and individualized treatment pathways.
20. Tagliafico, A.S., Piana, M., Schenone, D., Lai, R., Massone, A.M., Houssami, N. Overview of radiomics in breast cancer diagnosis and prognostication. Breast 2020, 49, 74-80, https://doi.org/
doi: 10.1016/j.breast.2019.10.018.21. Branco PESC, Franco AHS, de Oliveira AP, Carneiro IMC, de Carvalho LMC, de Souza JIN, Leandro DR, Cândido EB. Artificial intelligence in mammography: a systematic review of the external validation. Rev Bras Ginecol Obstet 2024,46, e-rbgo71. https://doi.org/ 10.61622/rbgo/2024rbgo71. eCollection 2024.
22. Jannatdoust P, Valizadeh P, Saeedi N, Valizadeh G, Mobarak HS, Rad HS, Gity M. Computer-Aided Detection (CADe) and Segmentation Methods for Breast Cancer Using Magnetic Resonance Imaging (MRI). J Magn Reson Imaging 2025, https://doi.org/10.1002/jmri.29687.
We believe this will be sufficient at this time.
Comment 2
"My minor suggestion regarding the article is to share more up-to-date GLOBOCAN data."
Thank you again for your comment.
Of course, we apologize for outdated reference.
We included a newest report on cancer statistics from GLOBOCAN/IARC/WHO at the Reference no. 1.:
Bray F, Laversanne M, Sung H, et al. Global cancer statistics 2022: GLOBOCAN estimates of incidence and mortality worldwide for 36 cancers in 185 countries. CA Cancer J Clin. 2024;74(3):229-263. doi:10.3322/caac.21834
Yours sincerely
Team of Authors
Reviewer 2 Report
Comments and Suggestions for Authors
The topic of the article is interesting.
However, the presentation of the material needs significant revision.
The author provides a lot of descriptive information without quantitative indicators. Most of the information is subjective, which does not meet the requirements of the journal.
The author should provide the results of the study in quantity.
The text provided would be a good introduction and analysis of the approaches, however, it is still necessary to add the results of the studies in a more "scientific" form.
Therefore, it is possible to consider this article in more detail only if there is an objective presentation of the results of the study
Author Response
Comments:
"The author provides a lot of descriptive information without quantitative indicators. Most of the information is subjective, which does not meet the requirements of the journal.
The author should provide the results of the study in quantity.
The text provided would be a good introduction and analysis of the approaches, however, it is still necessary to add the results of the studies in a more "scientific" form."
We must draw attention to the fact that the article was conceived as an essay, and submitted to such a category. Therefore, in our opinion, the requirements for scientific method do not apply to it.
However, in view of the need for more scientific scrutiny, we have made several corrections, especially in the introduction and discussion sections and concerning the discussed case at the end.
Response:
Thank you for your comments.
However, we must disagree with these comments, because the work we sent is not an original research article, but an essay (as it was submitted), and should not be subject to the rigors of a strict scientific paper, as the Reviewer suggests.
However, in light of the need for greater scientific scrutiny, we have made some corrections, particularly in the discussion and case section at the end of the article.
We hope that the arguments we have provided are acceptable.
Round 2
Reviewer 2 Report
Comments and Suggestions for Authors
The authors made some edits, which allowed to improve the quality of the publication.
The publication in the form of an essay has a formed form and is eligible for publication.
However, it is worth paying attention to the number of references. If the publication is descriptive, then it is worth providing more references to existing publications to support the presented opinions.
Author Response
Thank you for agreeing for review our manuscript.
"However, it is worth paying attention to the number of references. If the publication is descriptive, then it is worth providing more references to existing publications to support the presented opinions."
Following your suggestion, we decided to expand the discussion a bit by adding references in places where they may be missing. We hope that this will be enough for the final acceptance of the text.